# Fungi in the Gut Microbiota: Interactions, Homeostasis, and Host Physiology

**DOI:** 10.3390/microorganisms13010070

**Published:** 2025-01-02

**Authors:** Hao-Yu Liu, Shicheng Li, Kennedy Jerry Ogamune, Abdelkareem A. Ahmed, In Ho Kim, Yunzeng Zhang, Demin Cai

**Affiliations:** 1College of Animal Science and Technology, Yangzhou University, Yangzhou 225009, China; haoyu.liu@yzu.edu.cn (H.-Y.L.); scli0302@outlook.com (S.L.); ogamune30@gmail.com (K.J.O.); 2Jiangsu Key Laboratory of Animal Genetic Breeding and Molecular Design, Yangzhou University, Yangzhou 225009, China; 3Joint International Research Laboratory of Agricultural & Agri-Product Safety, The Ministry of Education of China, Yangzhou University, Yangzhou 225009, China; 4Department of Veterinary Science, Botswana University of Agriculture and Natural Resources, Private Bag 0027, Gaborone P.O. Box 100, Botswana; aabdallah@buan.ac.bw; 5Department of Animal Resource & Science, Dankook University, 119 Dandero, Donnamgu Cheonan, Cheonan-si 31116, Republic of Korea; inhokim@dankook.ac.kr; 6Jiangsu Co-Innovation Center for Prevention and Control of Important Animal Infectious Diseases and Zoonoses, Yangzhou University, Yangzhou 225009, China; yzzhang@yzu.edu.cn

**Keywords:** microbiota, intestinal mycobiota, fungal–bacterial interactions, antibiotics, weaning

## Abstract

The mammalian gastrointestinal tract is a stage for dynamic inter-kingdom interactions among bacteria, fungi, viruses, and protozoa, which collectively shape the gut micro-ecology and influence host physiology. Despite being a modest fraction, the fungal community, also referred to as mycobiota, represents a critical component of the gut microbiota. Emerging evidence suggests that fungi act as early colonizers of the intestine, exerting a lasting influence on gut development. Meanwhile, the composition of the mycobiota is influenced by multiple factors, with diet, nutrition, drug use (e.g., antimicrobials), and physical condition standing as primary drivers. During its establishment, the mycobiota forms both antagonistic and synergistic relationships with bacterial communities within the host. For instance, intestinal fungi can inhibit bacterial colonization by producing alcohol, while certain bacterial pathogens exploit fungal iron carriers to enhance their growth. However, the regulatory mechanisms governing these complex interactions remain poorly understood. In this review, we first introduce the methodologies for studying the microbiota, then address the significance of the mycobiota in the mammalian intestine, especially during weaning when all ‘primary drivers’ change, and, finally, discuss interactions between fungi and bacteria under various influencing factors. Our review aims to shed light on the complex inter-kingdom dynamics between fungi and bacteria in gut homeostasis and provide insights into how they can be better understood and managed to improve host health and disease outcomes.

## 1. Introduction

The gastrointestinal (GI) tract of mammals is home to a large number and high diversity of microorganisms, including bacteria, archaea, viruses, and fungi, which together build a complex micro-ecosystem and shape the host’s physiology [1], e.g., the commensal gut microbiota supports host metabolism [2], maintains intestinal barrier function [3], and modulates the immune system, among other functions [4]. For instance, Lim and co-workers have demonstrated that *Lactobacillus sakei* OK67 can ameliorate blood glucose intolerance and obesity in mice fed a high-fat diet [5]. The anaerobic bacteria-derived butyrate has been shown to improve intestinal barrier integrity by increasing the expression of claudin-2 tight junction protein (TJ) through the IL-10 receptor α subunit (IL-10RA) [6]. Furthermore, commensal *Lactobacillus johnsonii* species can regulate the intestinal immunity of piglets by increasing the secretory immunoglobulin levels (SIgA) levels and improving T cell immunity [7]. In the past few decades, most of the studies have focused on the role of bacteria in the gut microbiome while overlooking the intestinal fungi, partly because of the low abundance of the latter, as they account for only 0.01–0.1% of the whole community [8,9]. However, accumulating evidence suggests that the fungal community, also referred to as the mycobiota, is indeed an important member of the microbiota, thereby requiring detailed characterization [10,11]. Despite large individual variations, mammalian intestinal fungi generally belong to the phyla Ascomycota, Basidiomycota and Chytridiomycota. The dominant groups include *Aspergillaceae*, *Nectriaceae* and *Trichocomaceae* families, as well as the three most abundant genera, *Candida*, *Saccharomyces*, and *Cladosporium* [12,13].

Under physiological conditions, fungi, bacteria, and viruses in the gut maintain a stable yet dynamic relationship with each other that can be neutral, synergistic, or antagonistic [14]. For example, the common fungus *Candida albicans* interacts with various bacteria, including *Clostridium difficile* and *Enterococcus faecalis* [15,16], altering their assembly and function through cell membrane contact, competition or cooperation for nutrients, and the production of secondary metabolites and antimicrobial peptides [17,18,19]. Numerous factors can affect these interactions and shape the gut microbiota, causing instability or stability [20,21,22,23], especially during early life stages [24]. Unlike in adults, the ‘young’ microbiota shows a higher degree of dynamism, with the composition and diversity of intestinal mycobiota being greatly influenced by environmental factors [25]. That being said, the gut mycobiota has also been implicated in various pathologies, particularly in intestinal inflammation and infection. Indeed, fungal dysbiosis has been linked to inflammatory bowel disease (IBD) in humans [26], obesity in mice models [27], and *Candida tropicalis*-induced diarrhea in piglets [28], etc. Nevertheless, it is plausible that the role of the mycobiota is context-specific, and the regulatory mechanisms of fungal–bacterial interactions remain to be explored.

To date, the exploration of gut microbiota has been hindered by traditional culture-dependent methods that can only identify and classify a small number of intestinal fungal species [24]. On the other hand, DNA-based culture-independent methods have been developed over the last decade, ranging from denaturing gradient gel electrophoresis [29], restriction fragment length polymorphism analysis, and the oligonucleotide fingerprinting of rRNA genes [30], to high-throughput sequencing techniques such as 18S ribosomal DNA and internal transcribed spacer regions (ITS) [31] as well as state-of-the-art third generation of sequencing, also known as long-read sequencing, which offers in-depth analysis and renews our understanding of the composition and diversity of fungal species despite having its own drawbacks [32]. Hence, this review will first outline the methodologies for studying microbiota including both culture-dependent and culture-independent methods. Secondly, we will summarize the knowledge about gut microbiota colonization and composition in mammals with an emphasis on their early life stages. Most importantly, we will discuss the significant role of intestinal mycobiota in micro-ecosystems, highlighting the potential effects of fungal–bacterial interactions in host gut homeostasis.

## 2. Methods for Studying Intestinal Mycobiota

Previously, research on microbiota mainly relied on culture-based methods, which greatly restricted the in-depth analysis of community composition, particularly for minor components like fungi and in disease situations where microbiota/mycobiota homeostasis is disrupted [33,34]. This issue has largely been resolved thanks to rapid advancements in culture-independent methodologies, such as next-generation sequencing (NGS) technology and bioinformatics analysis. These methodologies include 18S ribosomal DNA (18S rDNA) and ITS [35], which are also integrated with various approaches such as enriched culturing, immunofluorescence staining and imaging, eukaryotic and prokaryotic cell flow cytometry [36], and metagenomics [33]. Each technique has its own set of strengths and limitations, requiring the careful consideration of the research objectives and the specific mycobiota being targeted (summarized in Table 1).

### 2.1. Culture-Dependent Methods

Previously, techniques reliant on cultural media have persevered as the primary means of observing the multifariousness and morphological framework of the intestinal microbiota [51]. Their efficacy and accuracy are closely linked to the cultivation matrix. For example, Sabouraud dextrose agar is commonly used for filamentous fungi cultivation due to its low pH and incubation temperature of 25 °C [38]. Blood agar (BA) and chocolate agar (CA) can support the growth of fungi associated with fungal keratitis [41]. Blood agar and CA typically contain blood components and are used for the isolation and cultivation of aerobic bacteria such as *Neisseria* and *Haemophilus* [39]. However, the presence of blood components can lead to excessive bacterial growth, affecting the isolation efficiency, and improper technical handling can easily cause contamination [40]. The ID Fungi Plate (IDFP) is a novel culture-dependent method designed to improve the identification of filamentous fungi and bacteria using matrix-assisted laser desorption ionization time-of-flight mass spectrometry (MALDI-TOF MS) [52]. It has become a potent tool for microbial identification and diagnosis, offering a method that is faster, more sensitive, and cost-effective, enabling the rapid identification of anaerobic bacteria and other bacteria that are difficult to culture [42]. However, the application of MALDI-TOF MS also has certain limitations. It requires high purity and many processing steps, making it difficult to identify complex samples. In addition, the identification results depend on existing databases [43]. Many fungi do grow on standard media, making culture-dependent methods the most direct and feasible way to visualize the colony morphology and community color, and they are recognized as the “Golden standard” [24]. However, some fungi found in mammals require very specific media conditions. For example, *Malassezia* species fail to grow in the absence of fatty acids [53]. The majority of fungi from ruminants require an anaerobic environment and specific additions such as wheat straw for culture [54]. Furthermore, with culture-dependent methods, it might be difficult to distinguish between species and genera of some groups of fungi that are morphologically similar, and some fungi cannot be cultured with the existing methods [55]. Interestingly, the development of culturomics has led to the renaissance of culture-dependent methods. Borges et al. analyzed the diversity of gut fungi in humans from eutrophic, overweight, and obese individuals utilizing different culture histology methods, revealing significant differences within specific filamentous fungal groups [56].

### 2.2. Culture-Independent Methods

To overcome the limitations of traditional culture techniques, the latest deep sequencing techniques based on whole genome analysis (WGS) and bioinformatics analysis have been developed and implemented [57,58]. These include 18S rDNA, ITS [49], and metagenomics [33] among others. In eukaryotes, 18S and 28S rRNA genes have been used to characterize the entire eukaryotic community in an environment [9]. Located between the 18S, 5.8S, and 28S rRNA genes are internal transcribed spacers, called ITS1 and ITS2, respectively, and these regions are present exclusively in fungi [59]. The technical and bioinformatic requirements for mycobiome sequencing are comprehensively discussed elsewhere [59]. In brief, amplicon sequencing employs fungal-specific primers to amplify hypervariable domains within the ITS or 18S regions of the rRNA gene locus [11]. Although the ITS region is highly variable, it may not be able to distinguish differences at the subspecies or strain level in some cases, and the complexity of the ITS region can lead to difficulties in sequencing and assembly, requiring high-quality samples [49,50].

Moreover, NGS technology, such as metagenomics, enables the analysis of community genomes within specific environments [60]. The exploration of microbial community diversity and composition lays a solid foundation for delving deeper into the intricate connections among microbial community functionality, inter-cooperative dynamics, and the encompassing environment [60,61]. Furthermore, metagenomic studies can be used to infer associations between the gut microbiome and the host phenotypes. For example, Chen et al. constructed a reference gene catalog of the pig gut microbiome using fecal samples, known as the Porcine Integrated Gene Catalog, through deep metagenomic sequencing [47,62]. They identified differences in strains of bacteria such as *Bacteroides* and *Bifidobacterium* between wild boars and Duroc pigs, which expanded resources for further research on the pig gut microbiome [47]. However, in the current landscape of fungal genomics, sequencing technologies exhibit potential biases, necessitating the careful consideration of sample collection and experimental data analysis [45]. The accurate identification of certain fungal sequences poses challenges due to the insufficient availability of fungal genomes [46].

The rapidly evolving field of intestinal microbiome research increasingly relies on metagenomic sequencing and other omics approaches. However, there is no universal method to characterize the entire gut microbiome and the availability and handling of big data will be vital, while the interpretation of collected data may be method-specific [63]. By employing new methods such as meta-transcriptome and metaproteome analysis, our understanding of the intestinal microbiome and its impact on health and disease will be unlocked [64].

## 3. Intestinal Colonization of Fungi and Bacteria

Immediately after birth, the microbiota begins to form an important layer of the intestinal barrier, providing colonization resistance against pathogens [65,66]. Notably, the ingestion of colostrum and milk appears to be one of the most important factors determining the trajectory of microbial succession in neonates, as it provides essential nutrients and bioactive compounds [67,68,69]. Other sources including diet, maternal influences, and environmental factors (e.g., season, geographical location, and hygiene level) contribute to gut microbiota establishment during the animal’s development into adulthood [43,70], indirectly impacting the maturity of the host’s immune system and its functionality [71].

### 3.1. Early Colonization of the Intestinal Microbiota

It was once suggested that fungi may colonize the host at birth or even prior to it. However, this has yet to be definitively determined due to the inherent limitations of sampling and detection methods [72]. Nevertheless, the succession of intestinal bacteria and fungi begins immediately after birth. In livestock science, studies on the economically important species, such as pigs, often focus on gut microbiota development during weaning [73]. Events in the gut during this crucial period of time drive the long-term growth and immune capabilities of the animals. Meanwhile, it presents a strategic opportunity to manipulate the maturation and functionality of the gut microbiota in pigs. Such insights could also enhance our understanding of the gut microbiota developmental trajectories in human infants [74,75].

The microbial colonization process is profoundly impacted by both maternal and environmental factors [76,77], which is confirmed by the shared fecal microbial communities between suckling piglets and their mothers [76]. In commercial farming, the weaning age of piglets commonly falls between 21 and 35 days. Thereafter, the piglets undergo a drastic dietary shift from liquid milk to solid feed, which has a relatively high content of protein and dietary fiber [78]. This abrupt dietary change triggers a considerable weaning stress response and has consequential effects on the physiological and psychological state of the piglets, leading to substantial changes in gut microbiota composition and great immune responses [79,80]. This process, called the ‘weaning reaction’, is not pathological but necessary, as animals without microbiota colonization are proven to be more susceptible to inflammation and infection [81,82]. From birth to post-weaning, the gut microbiota of piglets undergoes drastic fluctuations: Upon birth, the prevailing resident bacterial genera are *Bacteroides*, *Escherichia*, and *Clostridium* [83]. As time progresses, the prevalence of *Prevotella* increases, supplanting *Bacteroides* as the third-most dominant genus as early as the piglets’ second week of life [84]. It is suggested that there is a strong correlation between the surge in *Prevotella* abundance and the increased consumption of carbohydrates over an extended duration, resulting in its dominance after weaning [83,85]. Furthermore, the abundance of *Blautia* [86], *Oscillibacter* [87], *Roseburia* [87], and *Oscillospira* [87] also increases during the transition from birth to weaning in the gut of piglets. In contrast, the prevalence of genera such as *Bacteroides* [83], *Parabacteroides* [88], *Lactobacillus* [80], and *Clostridium* [80] declines. These changes together result in an increase in the richness and diversity of the gut microbiota as swine advance in age [89], similar to what is observed in humans and other mammals [24,90].

Intestinal fungal colonization starts with *Candida*, *Malassezia*, *Cladosporium*, and *Debaryomyces* as the primary colonizers, which then shifts to *Saccharomyces* dominance at weaning [91]. Next-generation sequencing analysis reveals that Ascomycota and Basidiomycota are the most prevalent fungal phyla in piglet feces [92]. During the pre-weaning period, the predominant fungal families in piglets include *Trichosporonaceae*, *Symbiotraphinaceae*, *Mucoraceae*, and *Cladosporaceae*, which are also present in the soil and environment [93]. During the weaning period, the fungal content in piglet feces increases compared to birth and becomes more pronounced at the end of weaning [94], primarily consisting of *Mucor*, *Cladosporium*, and *Trichosporon* families [95]. Meanwhile, the microbial landscape of post-weaning piglets undergoes a significant shift two weeks after weaning. With the emergence of *Dipodascaceae* and *Aspergillaceae*, the previously dominant family is supplanted by *Saccharomycetaceae*, marking a notable and enduring transformation in the gut mycobiota [92,95]. In humans, *Debaryomyces hansenii* is predominant during the initial months, while *Saccharomyces cerevisiae* is the most abundant fungal species in the gut of infants aged 1 to 2 years [96].

### 3.2. Distribution of Intestinal Mycobiota

In most monogastric animals, fungi are present in the GI tract [12], where their abundance gradually increases from the ileum to the colon under normal conditions, reaching its highest level in the distal colon [97]. In humans, the predominant phyla are identified as Ascomycota, followed by Basidiomycota and Zygomycota [9,11,98], while the 10 genera with the highest abundance include *Candida*, *Saccharomyces*, *Penicillium*, *Aspergillus*, *Cryptococcus*, *Malassezia*, *Cladosporium*, *Galactomyces*, *Debaryomyces* and *Trichosporon* [99]. In the small intestine of healthy individuals, the major fungal genera include *Candida*, *Galactomyces*, *Malassezia*, and *Saccharomyces* [100,101], while the major fungal genera in the colon are *Candida*, *Malassezia*, *Cladosporium*, *Fusarium*, *Galactomyces*, *Pichia* and *Phaeococcomyces* [102,103]. In the murine intestine, commonly found genera such as *Aspergillus*, *Candida*, and *Cryptococcus* have also been identified, though in limited numbers [12].

In pigs, the major fungal phyla Basidiomycota, Ascomycota, Chytridiomycota, Zygomycota, Glomeromycota, and Neocallimastigomycota are distributed differently along the GI tract [22]. The phyla Ascomycota and Basidiomycota are the most prevalent within the fungal kingdom, situated at the apex of the fungal evolutionary tree, with approximately 98% of known fungal species belonging to these two phyla [94,104]. Interestingly, it was found that *Kazachstania slooffiae* is the predominant fungal species in weaned piglets, but it is diminished in humans and mice [94,105]. In the porcine ileum, the main fungi at the phylum level are Ascomycota, Zygomycota and Basidiomycota, while at the genus level, fungi include *Lysurus*, *Microidium*, *Phallus*, and *Cladosporium* [106,107]. The relative abundance of Ascomycota and Basidiomycota in the colon of weaned piglets, which are positively correlated with each other, is significantly higher than in the cecum [22,108]. Indeed, the cumulative abundance of the phyla Ascomycota and Basidiomycota in the colon constitute a substantial 80% of the overall fungal population [104,108,109]. Among all identified fungal genera, *Saccharomycopsis*, *Wallemia*, *Bifiguratus*, *Russula*, *Bullera*, *Mrakia*, and *Kazachstania* are the most abundant genera in the cecal digesta of piglets [107,110], while *Candida*, *Vishniacozym*, and *Kazachstania* are the most abundant fungal genera in the growing-pig cecum [110]. Additionally, *Kazachstania*, *Saccharomycopsis*, *Aspergillus*, *Scheffersomyces*, and *Issatchenkia* are the dominant genera in the colon of pigs [92,107].

Notably, in contrast to the intestinal bacterial community, the relative proportions of different phylum levels within the intestinal fungal community exhibit limited variability across varying stages of piglet development [111,112]. Summers et al. showed that the diversity and composition of intestinal fungi in pigs’ feces remained relatively stable across sampling sites from day 1 to day 35 of a piglet’s life [94,113]. It is suggested that compared to the bacteriome, the intestinal fungal community exerts a more consistent interaction or influence on gut development during the critical early post-weaning period of the piglets, as it maintains a remarkable state of stability [84]. It has been discovered that the proportionate prevalence of *Aspergillus*, *Cladosporium*, *Simplicillium*, and *Candida* increases with the maturity of the piglets [104,110]. Nonetheless, the proportional representation of *K. slooffiae* and *Aureobasidium* demonstrates a notable decline as piglets age [105,114]. It is also worth noting that the relative prevalence of *Hanseniaspora* and *Penicillium* initially rises and then declines gradually over time [104,115]. Nonetheless, research on fungi is still in the preliminary exploratory stage, and new findings will greatly contribute to our understanding of the role of fungi in gut homeostasis in mammals [116].

## 4. Relationships Between Fungi and Bacteria in the Intestine

The role of gut bacteriome, including the commensal species, has been extensively studied, whereas our understanding of other microorganisms remains limited [26,117]. The gut mycobiota is another important microbial group and has been implicated in various host gut physiological events and energy metabolism, as well as in diseases [9,12]. Most studies on intestinal mycobiota have focused on humans and mice, with limited research into livestock species [104].

Recent studies have shown that fungi play an equally important role in maintaining homeostasis of the bacterial microbiota and regulating overall gut health [1,118]. Wheeler et al. demonstrated in a murine cutaneous infection model that *Malassezia* selectively induces IL-17, playing a crucial role in coordinating the host antifungal immune response [119]. Additionally, in mice, bacteria are found to prevent the overgrowth of *C. albicans*, while *C albicans* can also alter the bacterial microbiota, such as promoting the growth of *Bacteroides* but antagonizing *L*. *johnsonii* [120,121]. Meanwhile, several studies have demonstrated the clinical relevance of interactions between fungi and bacteria, such as in bloodstream infections. For instance, *Candida* species and *Pseudomonas* [122] are often co-isolated in cases of bacteremia and candidemia, which frequently result in significant morbidity and mortality, especially in children [122,123]. Furthermore, in immunocompromised patients, such as those with acquired immunodeficiency syndrome (AIDS) and inflammatory bowel disease (IBD), dysbiosis of the bacteriome can disrupt the mycobiome, leading to fungal infections and a worsening of disease progression. Sokol et al. [124] found that the unique inflammatory environment in IBD inhibits bacterial growth, resulting in an increased Basidiomycota/Ascomycota ratio and a higher abundance of *C. albicans* in the fecal samples of IBD patients compared to healthy subjects. In addition, microbes including *Escherichia coli*, *C. tropicalis*, and *Streptococcus constellatus*, which are abundant in IBD, have been shown to be proficient in biofilm formation, thereby promoting each other’s growth [125]. There are complex interrelationships between fungi and bacteria in health and disease, which can be categorized into three primary types: mutualism, commensalism, or competition. These interactions can be realized through host regulation, such as immunity, directly or indirectly [16].

### 4.1. Negative Interactions

Specific bacterial and fungal interactions can be used as tools to improve gut homeostasis. Intestinal fungi can restrict bacterial colonization through the production of alcohol, antimicrobial peptides, and some metabolites [126,127]. Certain fungi can selectively impact changes in the bacterial community [128]. For example, *S. boulardii* is a widely studied probiotic that shows protective effects against various bacterial pathogens, including *C. difficile*, *Helicobacter pylori*, *Vibrio cholerae*, *E. coli*, *Salmonella Typhimurium*, and *Shigella flexneri* [129,130,131,132]. Furthermore, baker’s yeast *S. cerevisiae* is one of the most studied and representative eukaryotes [133]. Despite reports suggesting that this yeast occasionally causes surface and systemic infections [134], a related study demonstrated that *S. cerevisiae* can functionally replace gut bacteria through mannans to protect both local and systemic immunity [135]. In addition, *S. cerevisiae* was found to reduce the translocation of the enterotoxin-producing enterotoxigenic *Escherichia coli* (ETEC), inhibit bacterial growth and colonization, and suppress ETEC adhesion [136]. Another example is *C. albicans*, which has been found to influence the reorganization of gut bacteria following antibiotic treatment [137]. In a study including 178 preterm infants (fecal samples at six weeks postpartum collected), it was inferred that *C. albicans* antagonized *Klebsiella pneumoniae* [18].

In turn, bacteria can regulate fungal development and mycelial growth by producing fatty acids, lactic acid, and butyrate [138]. In one study, *C. albicans* colonization in the murine gut was found to be restricted by Firmicutes (*Clostridium* clusters IV and XIVa) and Bacteroidetes [139]. The authors subsequently showed that these symbiotic anaerobes activated the expression of hypoxia-inducible factor (HIF)-1α and the antimicrobial peptide LL-37-CRAMP, which stimulated gut mucosal immune effectors to restrict fungal colonization. In a murine study, *Bacteroides thetaiotaomicron* and *L*. *johnsonii* were found to suppress the growth of *Candida* species, thus alleviating dextran sodium sulfate (DSS)-induced colitis [140]. The authors observed that these bacteria enhance the host’s anti-inflammatory responses through the upregulation of TLR9 and activation of chitinase-like protein-1, facilitating fungal clearance. Another example is *E. faecalis*, where in vitro studies have shown that it is antagonistic to *C. albicans* and reduces its virulence by affecting the filamentous structure [141]. Some bacteria exhibit significant antifungal behavior, such as *Serratia marcescens* (belonging to the *Yersiniaceae* family) which can use a type VI secretion system to deliver antifungal toxins that eliminate both the yeast and hyphal forms of *C. albicans* [142]. The ongoing exploration of negative interactions between intestinal fungi and bacteria unveils potential probiotics and associated intervention strategies for managing intestinal disorders.

### 4.2. Positive Interactions

Some interactions in the gut microbiota may have detrimental effects on gut homeostasis. *C. albicans* can exert a deleterious effect on the host by reducing dissolved oxygen in the vicinity of the bacteria, thereby promoting the growth of anaerobic bacteria, including *C. difficile* [143,144]. Furthermore, *C. albicans* may produce antioxidants that facilitate the growth of anaerobic bacteria in aerobic environments [145]. The presence of *C. albicans* exacerbates the severity of *C. difficile* infections in mice. Additionally, Santus et al. discovered that *Salmonella* can utilize fungal iron carriers in the intestine to acquire iron, thereby gaining a growth advantage [117]. These iron carriers can be produced by symbiotic fungi, through cross-feeding, or from diets containing fungal xenosiderophores [117,146]. The oral administration of β-lactam antibiotics in mice results in the proliferation of *C. albicans* hyphae in the gut, which may be attributed to the release of bacterial peptidoglycan subunits in the gut lumen [147].

Interactions between intestinal fungi and bacteria further influence host physiopathology. One study found that the elimination of the *Enterobacteriaceae* family of bacteria from the gut negated the beneficial effects of the yeast *S. boulardii*, suggesting that intestinal bacteria may mediate the effects of fungi on their hosts [148,149]. These studies underscore the collaborative role of intestinal fungi and bacteria in exacerbating fungal burden, promoting bacterial overgrowth, and subsequently influencing host gut homeostasis [19].

## 5. Environmental Factors Affecting the Fungi and Bacteria Interactions

The interaction between the gut mycobiome and bacteriome is an act of balance, influenced by various factors such as different life stages, nutrients, and drug use (Figure 1) [150,151,152]. In pigs, the early colonization and composition of fungi can be greatly impacted by diet (e.g., dietary component and pattern), nutrition [56], and antibiotic administration [67]. It has also been found that some intestinal diseases, such as diarrhea, can significantly impact the structure of the intestinal microbiota in piglets [153].

### 5.1. Antibiotics

The interaction between bacteria and fungi is best exemplified in animals treated with antibiotics. For decades, antibiotics have been widely used in the management of bacterial infections, disease prevention, the enhancement of animal wellness, and the promotion of animal growth performance [67]. The effect of antibiotics on the composition of the intestinal microbiota may vary due to factors such as differences in intestinal compartments and the metabolism of antibiotic drugs [154]. Nevertheless, antibiotic therapy has been demonstrated to exert a substantial bactericidal effect within the gut [155,156]. At the same time, it has been linked to the proliferation of certain fungi. The administration of antibiotics has been observed to modify the metabolic reservoir within the intestines in a manner that fosters *C. albicans* expansion [120,157]. On the other hand, antibiotics can affect host immunity by altering microbial metabolites and signals, particularly those recognized by intestinal epithelial and immune cells, this may affect the production of lipids, bile acids, amino acids, and amino acid derivatives in the gut, with changes in the mycobiota also playing a role [158].

In a study using piglets as a model to investigate the impact of early antibiotic intervention, the supplementation of therapeutic antibiotics from day 7 to day 42 postnatal was observed to induce distinct alterations in the microbial composition of both the small and large intestines [154]. The most significant changes occurred in the small intestine, with decreases in *Clostridium* and *Bacillus* in the stomach, duodenum, and jejunum, as well as decreases in *Prevotella* in the colonic digesta and increases in *Ehrlichia* found in the jejunum [159]. Antibiotic treatment impaired some commensal bacteria, resulting in reduced short-chain fatty acid (SCFA) production, lower levels of T helper 17 (Th17) and regulatory T (Treg cells), and increased intestinal inflammation [160]. In an experimental model involving 18-week-old swine, the oral application of the therapeutic antibiotic ASP250 (a composite of aureomycin, *sulfadimethoxine*, and penicillin) was observed to amplify *E. coli* strains, along with the presence of antibiotic resistance genes in fecal matter [96]. Moreover, ASP250 elevated the quantities of *E. coli* and *Lachnobacter* in the cecum and colon of growing swine [68]. Following three days of clindamycin administration, mice exhibited heightened proliferation of *Enterococci* and gram-negative bacteria, consequently leading to augmented colonization by *C. difficile* [161]. The impact of antibiotics on the bacterial microbiota is known to induce alterations that affect the microbial community structure; however, the influence of antibiotics on the commensal fungi within the microbiota remains not fully elucidated [162]. In contrast, it has been shown that antibiotic treatment has a more lasting effect on gut fungi than on bacteria [163]. Metagenomics and metatranscriptomics analysis of the gut mycobiome from healthy humans who received one of five antibiotics (doxycycline, azithromycin, augmentin, ciprofloxacin, and cefuroxime) revealed that over one-third of fungal species exhibited significant changes even 90 days after treatment. The community shifted from mutualism to competition, whereas the bacterial community was largely recovered [163]. In another study, the effects of amoxicillin and macrolides on the gut microbiota of 21 infants predisposed to respiratory syncytial virus (RSV) showed an increased relative abundance of the phylum Basidiomycota and a significantly higher relative abundance of the genus *Candida* more than 6 weeks after the start of treatment [164]. Several studies indicate that fungal burden increases during antibiotic therapy due to the release of bacterial ecological niches [120,165]. In contrast, Spatz et al. reported a reduction in overall fungal counts in murine feces following treatment with the broad-spectrum antibiotic amoxicillin–clavulanic acid [26]. Subsequent experiments identified *Enterobacter hormaechei* as potentially capable of diminishing fungal populations through competition for nutrients and intercellular adhesion [26,166]. The study shows that after antibiotic administration, the abundance of some common intestinal fungi, including *Saccharomyces* spp. and opportunistic pathogens such as *C. albicans*, *Candida parapsilosis*, and *Malassezia restricta*, decreased, while some less common fungi like *C. boidinii* increased in abundance [163]. Noverr et al. exposed mice to *Aspergillus fumigatus* and orally administered *C. albicans* followed by antibiotic treatment to construct a mouse model of microbiota disruption induced by antibiotics, thereby inducing the development of allergic airway disease [167]. The introduction of ecological dysbiosis in the gut through antifungal and anti-bacterial treatments is often used as an approach to study the interactions between fungi and microorganisms [165,168]. Numerous experiments have shown that broad-spectrum antibiotics and anaerobic-specific antibiotics can have differential effects on fungal susceptibility, predisposing patients to intestinal infections with *C. albicans* [169,170]. Usually, animals with healthy bacterial communities are significantly more resistant to disease-causing fungi (e.g., *C. albicans* colonization) than animals treated with antibiotics [152]. Studies have shown that *C. albicans* affects the recolonization of the intestinal microbiota in antibiotic-treated mice and the administration of antifungal drugs to mice treated with DSS-exacerbated colitis [118,171]. The impact of antibiotics on fungi is multifaceted, as the depletion of intestinal fungi with antifungal drugs may also trigger the growth of pathogenic bacterial microbiota. We propose that these interactions should be considered in clinical settings where such drugs are used, as well as in animal production where antibiotic abuse is widespread.

### 5.2. Diet and Nutrition

Many residents of our gut microbiome originate from food/feed and are therefore greatly impacted by diet and nutrition. For instance, after providing healthy volunteers with diets rich in different fungi, such as *Penicillium* (blue cheese), *Saccharomyces* (multiple foods), and *Phomaherbarum* (beer), the corresponding fungi from the diets were detectable in fecal samples [66]. Indeed, it has been proposed that the source of oral fungi and diet can explain all the fungi contained in the feces [172]. In addition to specific fungi components included in diets, dietary patterns, microbiota-derived metabolites, and dietary contamination, such as mycotoxin, all strongly influence the composition of the intestinal mycobiota [173,174].

The components of diet are some of the most important factors in altering the gut microbiota, including carbohydrates, proteins, and fats [175]. Carbohydrates are the primary source of energy for humans and most single-stomach animals, accounting for 60 to 70 percent of total energy intake [176]. The potential relationship between the ratio of fungi and bacteria is observable, with a higher prevalence of *Prevotella* in individuals who maintain a high-carbohydrate diet, while *Candida* and other fungi may exhibit positive associations with short-term carbohydrate intake [177,178]. Some low-abundance fungal genera, such as *Saccharomycopsis*, *Mrakia*, *Wallemia*, *Cantharellus*, *Eurotium*, *Solicoccozyma*, and *Penicillium*, are associated with glucose and fructose concentrations and also play a role in the degradation of dietary carbohydrates, as shown in the porcine colon [92]. Amino acids and proteins can have varying impacts on the abundance of *C. albicans* in in vivo and in vitro models. In vivo, the abundance of *C. albicans* displayed a negative correlation with amino acid content [179]. In contrast, experiments conducted using in vitro models demonstrated a positive correlation between *C. albicans* and amino acid uptake [180], likely due to the fungus converting amino acids into carbohydrates. Gut fungi are also influenced by high-fat diets. In murine models, a high-fat diet has been associated with a diminished prevalence of certain fungal genera, including *Alternaria*, *Saccharomyces*, *Septoriella*, and *Tilletiopsis* [10]. Dietary components are complex and nutrients interact with each other, making it difficult to study their precise effects on gut fungi, necessitating deeper exploration.

Dietary patterns mainly refer to changes in dietary habits, for instance, weaning piglets transitioning from milk to solid feed [104], or shifts between animal-based and plant-based diets [181], all of which can affect gut fungi. It has been shown that the transition from breast milk to a solid diet before and after weaning is the main factor determining the alteration of gut mycobiota composition and structure in piglets [182]. Furthermore, the structure and metabolic pathways of the intestinal fungi in piglets fed with milk replacer differed significantly from those fed with breast milk, which may increase the risk of intestinal diseases in the former group [183,184]. In healthy individuals, a plant-based diet was associated with increased intestinal colonization by *Candida* species, whereas an animal-based diet promoted colonization by *Penicillium* species [23].

In addition, the gut microbiota can exert influence not only by directly interacting with other microorganisms but also indirectly through microbiota-derived metabolites, such as SCFAs, tryptophan, and bile acid products [185]. Short-chain fatty acids, mainly butyrate, acetate, and propionate, have been shown to play a significant role in microbial functions [186,187]. For example, it is well known that butyrate is the main source of carbon energy for colonocytes [188,189]. It has been reported that hindgut fungal species in pigs are mainly associated with acetate concentrations [190]. *Metschnikowia* and *Tomentella* are the two fungal genera that show strong correlations with the production of acetate in the pig intestine, suggesting that pig intestinal fungi may be involved in the metabolism of polysaccharides in diets [191]. In contrast, some fungal genera with relatively low abundance (approximately 0.1% of total operational taxonomic units, OTU), such as *Tomentella*, may be the only fungal genus associated with butyrate production in the pig intestine, suggesting its potential role in maintaining gut homeostasis [22].

Exposure to mycotoxin contamination in feed increases the risk of poisoning in both humans and livestock, particularly in pigs [192]. Mycotoxins are secondary metabolites produced by fungi, which possess toxicity and can damage the intestine, liver, and multiple other organs, as well as affect the immune system of humans and animals [193,194,195]. Currently, the most concerned mycotoxins include aflatoxin B1 (AFB1), ochratoxin A (OTA), fumonisin B1 (FB1), zearalenone (ZEN), and deoxynivalenol (DON) [196]. Remarkably, these mycotoxins also affect the colonization and growth of intestinal fungi. It is reported that ZEN can significantly inhibit the biofilm formation and hyphal development of *C. albicans* in in vitro culture experiments [197]. Furthermore, enniatins (ENNs), which are another type of mycotoxin produced by *Fusarium*, have been proven to exhibit certain antifungal activity against probiotic yeasts *S. cerevisiae* [198].

### 5.3. Other Factors

In addition to the main factors mentioned above, there are a variety of other factors that can influence the composition of the gut microbiota, such as gestational age [199]. Prematurity may severely affect the development of the gut and the systemic immune system as the colonization of the gut microbiota in preterm infants is threatened by immature organ development and environmental factors after birth [200]. Studies on the gut microbiota composition in preterm pigs showed that *Enterococcus*, *Escherichia*, and *Clostridium* dominated their intestines [201]. Seasonal changes also alter the gut microbiota. In particular, the alpha diversity of the gut fungi in wild-living Tibetan macaques (*Macaca thibetana*) is found to be highest in winter, followed by autumn, spring, and summer [202].

For both humans and livestock, the impact of physical conditions (such as health or illness) on the composition of the host’s gut microbiota is very important [203,204]. For example, in IBD patients, microbial communities are typically perturbed, with a decreased abundance of Firmicutes and Bacteroidetes, and an increased proportion of gammaproteobacteria [203]. Diarrhea has been a global challenge for the livestock industry [205], affecting the growth of early-weaned piglets, damaging their intestinal tissues, and, in severe cases, leading to death [206], with gut microbiota dysbiosis playing a major role in its pathology [153]. Reports have shown that the diversity of gut microbiota in diarrhea-susceptible piglets is significantly reduced, with a decreased abundance of *Lactobacillus* and *Enterococcus* [207]. In recent years, research has found that diarrhea also alters the fungal community in the intestine [28]. Compared to healthy piglets, the ratio of *Derxomyces*, *Lecanicillium*, *Tuber*, and *Naganishia* is significantly reduced, while the abundance of *Kazachstania* and *Cortinarius* is increased in diarrheal animals [166]. Moreover, *K. slooffiae*, a crucial member of the pig fungal community, particularly in post-weaning fungi [208], is found to be more prevalent in piglets with diarrhea [108]. Meanwhile, *C. tropicalis*, one of the most abundant commensal fungi in the gut, has been shown to decrease in abundance in diarrheal piglets [12,28]. These studies indicate that diarrhea alters the intestinal microbiota of pigs, leading to dysbiosis and posing health threats in pig farming.

Notably, none of these factors exist independently of the other. For instance, the season and environment affect the foods/feeds on which mammals depend [202]. The environment also affects the host’s exposure to potentially pathogenic and non-pathogenic microorganisms or toxins [209]. In infants between 10 days and 3 months of age, *D. hansenii* from breast milk is the dominant fungal species [96]. However, as their diet changes from breast milk to solid food, the intestinal fungal communities shift from *D. hansenii* to *S. cerevisiae* [210]. Consequently, these factors are interlinked and interdependent and cannot be considered separately but need to be viewed as a whole; the same applies to the perception of the mycobiome and bacteriome.

## 6. Conclusions

In summary, the intricate interplay between the gut mycobiome and bacteriome represents an underappreciated yet significant field of exploration, relevant to both human and animal health as well as disease. Currently, there are numerous descriptive studies on the gut mycobiome and microbiome, but the in-depth exploration of interaction mechanisms is limited. It is evident that the fungal mycobiome, like the bacteriome, is influenced by a variety of factors, among which diet remains a major topic of research. Future studies should focus on the fine-tuning of the mycobiome and its interplay with the microbiome to foster a more comprehensive understanding of the gut mycobiome’s contribution to overall health and disease. It would be interesting to see how future research could combine multi-omics approaches, such as metagenomic analysis of the mycobiome with artificial intelligence (AI)-based technologies like machine learning to solve more mysteries in the field.

## Figures and Tables

**Figure 1 microorganisms-13-00070-f001:**
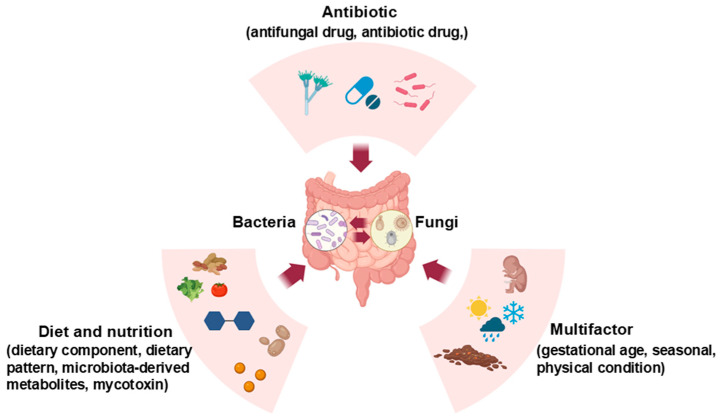
Environmental factors shape the interaction of mycobiota and microbiota in gut homeostasis.

**Table 1 microorganisms-13-00070-t001:** Methods for studying intestinal microbiota, pros and cons.

Type	Methods	Advantages	Disadvantages	Reference
Culture-dependent methods	SDA	1. Cultivates fungi under specific environmental conditions, such as filamentous fungi (low pH and temperature at 25 °C) 2. Used for the isolation of fungi	1. Fungal isolation efficiency is low 2. It cannot effectively inhibit bacterial growth	[37,38]
	BA and CA	1. Supports the growth of all fungi associated with fungal keratitis 2. Contains blood components 3. Used for the isolation and cultivation of aerobic bacteria, such as *Neisseria* and *Haemophilus*	1. Blood can lead to excessive bacterial growth 2. Vulnerable to contamination	[39,40,41]
	MALDI-TOF MS	1. Accurate, rapid, and economical 2. High sensitivity, high resolution, and wide applicability	1. High sample-purity requirements 2. Difficulty in identifying complex samples 3. Database update requirement	[24,42,43]
Culture-independent methods	Metagenomics	1. In-depth study of gut microbial diversity and composition 2. Infer associations between gut microbiome and host phenotypes, diseases	1. Complex sample handling and data analysis 2. Difficulty in identifying specific fungi that lack sufficient genomes	[44,45,46,47]
	18S rDNA/ITS	1. 18S and ITS regions can be used for species-level identification and classification 2. Study fungi that are difficult to cultivate	1. Cannot distinguish differences at the subspecies or strain level in some cases 2. Complexity of ITS region can lead to difficulties in sequencing and assembly, requiring high-quality samples.	[9,48,49,50]

## Data Availability

Not applicable.

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
