# Peer review of "Fungi in the Gut Microbiota: Interactions, Homeostasis, and Host Physiology"

_microorganisms, 2025, doi:10.3390/microorganisms13010070_

Round 1
Reviewer 1 Report
Comments and Suggestions for Authors
The authors write a review regarding the fungi in the gut microbiota of mammalian which present synthetically the pieces of information found in connection with ''Methods for studying intestinal mycobiota'', ''Intestinal colonization of fungi and bacteria'', ''Relation between fungi and bacteria in the intestine''; '' Environmental factors affecting fungi and bacteria interaction''. The manuscript is well written, and the information provided is correct and presented synthetically. I think this is the first review that treats a subject relating to the presence of other fungi in the mammalian microbiota, except the C. albicans and S.cereviasae, and this can represent the novelty of this manuscript, and for this reason, I recommend its publication. However, some corrections need to be made in the manuscript, as follows:
1) All scientific names of the microorganisms must be written in italics
2) A sentence cannot begin with an abbreviation. Authors must read with the attention their manuscript and rewrite these sentences ( see the sentences from R111; R 117; R 466 etc).
3) In the manuscript more abbreviations used are not explained. These explanations must be provided in the text, in the brackets, the first time when the abbreviation is used. Or, authors can add at the end of the manuscript a chapter entitled Abbreviations, in which all abbreviations used in the manuscript will be explained;
4) The bibliography from the Chapter entitled ''References'' is not written according to MDP rules. Authors must write all references according to the MDPI rules (authors can find these rules here: https://www.mdpi.com/journal/microorganisms/instructions#references).
Author Response
Dear editor,
Thank you for providing us with the opportunity to revise our manuscript for ease of comprehension and readability. We sincerely appreciate the time and effort you and the reviewers have dedicated to providing valuable feedback, which will undoubtedly help us improve the manuscript and ensure its quality for publication. We have worked on our manuscript following the reviewers’ advice, prepared a revised version highlighting all changes as red fonts and included point-by-point response letter as below.
Responses to Reviewer 1
Comments and Suggestions for Authors
The authors write a review regarding the fungi in the gut microbiota of mammalian which present synthetically the pieces of information found in connection with ''Methods for studying intestinal mycobiota'', ''Intestinal colonization of fungi and bacteria'', ''Relation between fungi and bacteria in the intestine''; '' Environmental factors affecting fungi and bacteria interaction''. The manuscript is well written, and the information provided is correct and presented synthetically. I think this is the first review that treats a subject relating to the presence of other fungi in the mammalian microbiota, except the C. albicans and S.cereviasae, and this can represent the novelty of this manuscript, and for this reason, I recommend its publication. However, some corrections need to be made in the manuscript, as follows:
Reply: Thank you very much for your valuable feedback and thoughtful comments on our manuscript. We appreciate your recognition of the novelty of our review manuscript and the overall positive assessment is encouraging.
- All scientific names of the microorganisms must be written in italics
Reply: We thank the reviewer for the reminder of our inconsistency. In the revised version of the manuscript, all scientific names of fungi and bacteria have been checked and italicized accordingly (highlighted in red font).
[Ln 49] ‘‘Furthermore, commensal Lactobacillus johnsonii species can regulate the intestinal immunity….’’
[In table 1] ‘‘Used for the isolation and cultivation of aerobic bacteria,such as Neisseria and Haemophilus’’.
[Ln 130] ‘‘For example, Malassezia species fail to grow in the absence of fatty acids [53]’’.
[Ln 164] ‘‘They identified differences in strains of bacteria such as Bacteroides and Bifidobacterium between wild boars and Duroc pigs, which expanded resources for further research on the pig gut microbiome [47]’’.
Others include, [Ln 346-348] ‘‘…C. albicans and Serratia marcescens’’, [Ln 367] ‘‘…Enterobacteriaceae family of bacteria…’’, [Ln 413] Enterococci and gram-negative bacteria, consequently leading to augmented colonization by C. difficile [161].’’ [Ln 443] Noverr et al. exposed mice to Aspergillus fumigatus…’’
- A sentence cannot begin with an abbreviation. Authors must read with the attention their manuscript and rewrite these sentences (see the sentences from R111; R 117; R 466 etc).
Reply: We appreciate the reviewer for the reminder, in the newest version, these changes have been made accordingly and highlighted in red font.
[Ln 114] ‘‘Blood agar (BA) and CA typically contain blood components and are used for the isolation and cultivation of aerobic bacteria such as Neisseria and Haemophilus [39]’’
[Ln 120] ‘‘It has become a potent tool for microbial identification and diagnosis, offering a method that is faster, more sensitive, and cost-effective, enabling rapid identification of anaerobic bacteria and other bacterial that are difficult to culture [42]’’.
[Ln 502] ‘‘Short-chain fatty acids, mainly butyrate, acetate, and propionate have been shown to play a significant role in microbial functions [187, 188]’’.
- In the manuscript more abbreviations used are not explained. These explanations must be provided in the text, in the brackets, the first time when the abbreviation is used. Or, authors can add at the end of the manuscript a chapter entitled Abbreviations, in which all abbreviations used in the manuscript will be explained;
Reply: We thank the reviewer for this valuable observation. In the revised manuscript, this has been revised and updated accordingly. In addition, we have updated our ‘Abbreviations’ section.
[Ln 50] ‘‘Furthermore, commensal Lactobacillus johnsonii species can regulate the intestinal immunity of piglets by increasing the secretory immunoglobulin (SIgA) levels and improving the T cell immunity [7]’’.
[Ln 328] ‘‘In addition, S. cerevisiae was found to reduce the translocation of the enterotoxin-producing enterotoxigenic Escherichia coli (ETEC), inhibit bacterial growth and colonization, and suppress ETEC adhesion [136]’’.
- The bibliography from the Chapter entitled ''References'' is not written according to MDP rules. Authors must write all references according to the MDPI rules (authors can find these rules here: https://www.mdpi.com/journal/microorganisms/instructions#references).
Reply: Thank you very much for your observation and recommendation. We have rewritten the references according to the MDPI rules as advised.
Reviewer 2 Report
Comments and Suggestions for Authors
I appreciate the opportunity to review the manuscript titled "Fungi in the Gut Microbiota: Interactions, Homeostasis, and Host Physiology." Below are my observations:
Integrate more critical discussion on fungal-bacterial interactions' clinical relevance, particularly their therapy implications.
On page 9, the authors mention that antibiotic treatment has a lasting effect on the fungal community (lines 394–396). This is a significant point, but it would benefit from including specific examples or studies to substantiate the claim.
In summary, this manuscript has substantial potential but requires revision for focus and clarity. I recommend resubmission following these improvements.
Author Response
Dear editor,
Thank you for providing us with the opportunity to revise our manuscript for ease of comprehension and readability. We sincerely appreciate the time and effort you and the reviewers have dedicated to providing valuable feedback, which will undoubtedly help us improve the manuscript and ensure its quality for publication. We have worked on our manuscript following the reviewers’ advice, prepared a revised version highlighting all changes as red fonts and included point-by-point response letter as below.
Responses to Reviewer 2
I appreciate the opportunity to review the manuscript titled "Fungi in the Gut Microbiota: Interactions, Homeostasis, and Host Physiology." Below are my observations:
Reply: We are truly thankful to the reviewer for taking the time to review our manuscript. And we value the insights and observations.
- Integrate more critical discussion on fungal-bacterial interactions' clinical relevance, particularly their therapy implications.
Reply: Thank you very much for your observation and recommendation. In the clean version of the revised manuscript, we have included some specific discussions on fungal-bacterial interactions to improve clarity shown as below and highlighted as red font in revised manuscript.
[Ln 298-311] ‘‘Meanwhile, several studies have demonstrated clinical relevance of interactions between fungi and bacteria, such as in bloodstream infections. For instance, Candida species and Pseudomonas [122] are often co-isolated in cases of bacteremia and candidemia, which frequently result in significant morbidity and mortality, especially in children[122,123]. Furthermore, in immunocompromised patients, such as those with acquired immunodeficiency syndrome (AIDS) and inflammatory bowel disease (IBD), dysbiosis of the bacteriome can disrupt the mycobiome, leading to fungal infections, and a worsening of disease progression. Sokol et al. [124] found that the unique inflammatory environment in IBD inhibits bacterial growth, resulting in an increased Basidiomycota/Ascomycota ratio and a higher abundance of C. albicans in the fecal samples of IBD patients compared to healthy subjects. In addition, microbes including Escherichia coli, C. tropicalis, and Streptococcus constellatus, which are abundant in IBD, have been shown to be proficient in biofilm formation, thereby promoting each other's growth’’.
On page 9, the authors mention that antibiotic treatment has a lasting effect on the fungal community (lines 394–396). This is a significant point, but it would benefit from including specific examples or studies to substantiate the claim.
Reply: Thank you very much for the constructive advice. In the revised manuscript, we have included more discussions on fungal-bacterial interactions to improve clarity and address the clinical relevance.
[Ln 418] ‘‘Metagenomics and metatranscriptomics analysis of the gut mycobiome from healthy humans who received one of five antibiotics (doxycycline, azithromycin, augmentin, ciprofloxacin, and cefuroxime) revealed that over one-third of the fungal species exhibited significant changes even 90 days after treatment. The community shifted from mutualism to competition, whereas the bacterial community largely recovered [164]. In another study, the effects of amoxicillin and macrolides on the gut microbiota of 21 infants predisposed to respiratory syncytial virus(RSV) showed an increased relative abundance of the phylum Basidiomycota and a significantly higher relative abundance of the genus Candida t [165]’’.
In summary, this manuscript has substantial potential but requires revision for focus and clarity. I recommend resubmission following these improvements.
Round 2
Reviewer 2 Report
Comments and Suggestions for Authors
Accept